# Fluid Biomarkers of Central Nervous System (CNS) Involvement in Myotonic Dystrophy Type 1 (DM1)

**DOI:** 10.3390/ijms24032204

**Published:** 2023-01-22

**Authors:** Salvatore Rossi, Gabriella Silvestri

**Affiliations:** 1Department of Neuroscience, Università Cattolica del Sacro Cuore–Sede di Roma, Largo F. Vito 1, 00168 Rome, Italy; 2Neurology Unit, Fondazione Policlinico Universitario Agostino Gemelli IRCCS, Largo A. Gemelli 8, 00168 Rome, Italy

**Keywords:** myotonic dystrophy type 1, DM1, CNS, biomarkers, central nervous system

## Abstract

Myotonic dystrophy type 1 (DM1), commonly known as Steinert’s disease (OMIM #160900), is the most common muscular dystrophy among adults, caused by an unstable expansion of a CTG trinucleotide repeat in the 3′ untranslated region (UTR) of *DMPK*. Besides skeletal muscle, central nervous system (CNS) involvement is one of the core manifestations of DM1, whose relevant cognitive, behavioral, and affective symptoms deeply affect quality of life of DM1 patients, and that, together with muscle and heart, may profoundly influence the global disease burden and overall prognosis. Therefore, CNS should be also included among the main targets for future therapeutic developments in DM1, and, in this regard, identifying a cost-effective, easily accessible, and sensitive diagnostic and monitoring biomarker of CNS involvement in DM1 represents a relevant issue to be addressed. In this mini review, we will discuss all the papers so far published exploring the usefulness of both cerebrospinal fluid (CSF) and blood-based biomarkers of CNS involvement in DM1. Globally, the results of these studies are quite consistent on the value of CSF and blood Neurofilament Light Chain (NfL) as a biomarker of CNS involvement, with less robust results regarding levels of tau protein or amyloid-beta.

## 1. Introduction

Myotonic dystrophy type 1 (DM1), also known as Steinert’s disease (OMIM #160900), is the most common muscular dystrophy in adults, with a prevalence of about 1:8000 among Caucasians. It is caused by an unstable expansion of a cytosine–thymine–guanine (CTG) trinucleotide repeat located at the 3′ untranslated region (UTR) of *DMPK* (Dystrophia Myotonica Protein Kinase) gene (OMIM #605377) in a pathological range from 50 to several thousand of repeats [1,2].

Mitotic and intergenerational instability of the pathological CTG expansion underlie the interindividual clinical variability and the anticipation phenomenon during parent-to-child transmission, respectively [3]. In DM1, the range of n(CTG) assessed in peripheral leukocytes inversely correlates with age at onset of symptoms, which in turn is associated with distinct severity of clinical presentation [4]: these include the most severe congenital form, with symptoms present at birth or even in late gestation, followed by the infantile (age at onset 1–10 years), juvenile (11–20 years), adult (21–40 years), and, finally, late-onset (>40 years) oligosymptomatic form [5].

Regarding individual DM1 manifestations, the n(CTG) in leukocytes directly correlates also with the global muscle disease severity [4], and it is associated with the development of restrictive respiratory syndrome [6], and severe cardiac conduction defects [7]. Results regarding the correlation between the n(CTG) expansion in leukocytes and other DM1 manifestations are controversial, likely because of somatic tissue mosaicism and the limited size of the cohorts assessed in different studies [8].

The pathological CTG expansion in *DMPK* produces damage or dysfunction in various DM1 tissues because of the nuclear accumulation of cytosine–uracil–guanine (CUG) expanded pre-mRNAs transcribed from the mutant *DMPK* allele, eventually affecting expression and/or activity of specific RNA-binding proteins, named muscleblind-like splicing regulator (MBNL) and CUGBP Elav-like family member 1 (CELF1), that act as developmental regulators of alternative splicing of many different genes in several tissues. The “spliceopathy” resulting from the toxic effect of expanded RNAs is at the basis of the typical multisystem involvement in DM1, which also affects, besides the skeletal muscle, the heart, the eye, the respiratory system, the endocrine system, and the central nervous system (CNS) [1,2,3].

CNS involvement is one of the core manifestations of DM1: cognitive, behavioral, and affective symptoms affect not only quality of life, but also prognosis of DM1 patients [9]. CNS manifestations vary based on clinical spectra of DM1, with an ongoing matter of debate being whether they might result from a neurodevelopmental, neurofunctional, or neurodegenerative defect, or their various co-occurrence [9].

In congenital onset DM1, mild-to-moderate intellectual disability and autism spectrum disorders are the main findings, whereas in infantile- and juvenile-onset forms, learning difficulties and attention deficits are often major issues. In the adult-onset DM1 form, a large proportion of patients manifests with cognitive defects that, however, may be overlooked by standard diagnostic tools, i.e., the Mini Mental State Examination (MMSE). Indeed, more extensive and sensitive cognitive batteries allow to reveal defects in frontotemporal functions, such as executive functioning, attention, visual memory, visual perception and construction, and social cognition [10]. Lastly, late-onset DM1 patients may present with exclusive or prevalent CNS manifestations in the form of frontotemporal dementia [11]. Besides cognitive abnormalities, the involvement of frontotemporal domains often manifests with significant mood and behavior disorders such as apathy, avoidant behavior, and misperception of the disease, which can significantly impact on personal, working, and social life of many DM1 patients [12].

Based on the most recent systematic review, concerning 41 studies on a total of 130 DM1 patients, the main neuropathological findings in DM1 brains consist of the presence of (1) protein and nucleotide deposits, (2) cellular alterations, and (3) white matter alterations [13]. Main deposits of neurofibrillary tangles (NFTs) composed of mis-spliced tau protein are found in the neurons from hippocampus and adjacent cortex, but without evidence of amyloid plaques in association, and eosinophilic ubiquitin-positive thalamic inclusions and Marinesco bodies in the substantia nigra have been also detected. Neuronal loss has been also inconsistently found in the brainstem, as well as in the cerebral cortex. Loss of myelin, dilation of perivascular spaces, gliosis, and capillary hyalinization in deep and subcortical white matter are the most common findings in DM1 brains, and it appears unlikely that there is a relation of these with cardiovascular risk factors [13]. Albeit similarities with other neurodegenerative diseases, all the neuropathological features described in DM1 patients are also seen, to a lesser extent, in normal aging, corroborating the idea that DM1 may represent a progeroid disease [14].

In the era of personalized medicine, and in view of forthcoming novel therapeutic perspectives of gene therapy supported in their efficacy by results of preclinical studies [15], CNS manifestations in DM1 should be always objectively quantified and monitored. This issue assumes particular relevance when we consider that CNS involvement in DM1 is undoubtedly prevalent, as it may indirectly affect patients’ prognosis by being at the basis of the poor insight regarding their disease condition and their scarce compliance to medical recommendations.

The availability of a specific biomarker can help to address this topic, allowing to objectively measure, and evaluate as indicator of normal biologic processes, pathogenic processes, or pharmacologic responses to a therapeutic intervention [16]. The use of a clinical biomarker is easier and less expensive than direct measurement of the final clinical endpoint, as it is usually measured over a shorter time span. Good biomarkers should be measurable with little or no variability, with a sizeable signal-to-noise ratio, and change promptly and reliably in response to changes in the system studied [16,17].

The slow progression of DM1 disease manifestations has prompted research to identify reliable and sensitive diagnostic, and possibly monitoring, biomarkers that are also specific for CNS involvement.

During the past years, many neuroimaging studies, such as conventional or high-resolution brain MRI, or functional studies assessing brain metabolism (18F-fluorodeoxyglucose positron emission tomography, proton magnetic resonance spectroscopy, single photon emission tomography) have been performed in DM1 [9]. Aims of these studies were to characterize the phenotypic spectrum of CNS alterations, to define their course over time, and to correlate them to the various DM1 clinical and molecular features. Moreover, several studies analyzed correlations between brain alterations and CNS clinical manifestations, particularly cognitive and behavioral symptoms, aiming to assess the value of brain neuroimaging as a biomarker of CNS severity and progression.

Among the still-unsolved issues pointed out by a specific, extensive, and updated review about this topic in DM1 [18], there was a need to clarify the role of neuroimaging studies as CNS monitoring biomarkers to be used in future therapeutic trials. In this regard, a few recently published longitudinal neuroradiological studies documented their capacity to detect only long-term gray and white matter disease progression in DM1 brains [19,20,21,22,23], suggesting that structural white matter (WM) and grey matter (GM) changes would progress very slowly in DM1 brains, so that they cannot represent a sensitive monitoring biomarker of disease progression. On the other hand, some of these studies support their reliability as diagnostic biomarkers of disease severity with regard to cognitive impairment, as specific structural WM changes correlate with specific cognitive visuospatial alterations [22].

The limited sensitivity of brain neuroimaging adds to DM1-related respiratory or cardiac comorbidities (i.e., orthopnea, previous implant of pacemaker and/or implantable cardioverter defibrillator), which might further affect DM1 patients’ compliance with longitudinal neuroimaging protocols. For these reasons, fluid biomarkers should represent a suitable alternative to be assessed in DM1 patients [24]. Since DM1 is mainly a spliceopathy, dysregulated expression of circulating micro-RNAs (miRNAs) has been assessed as a potential disease biomarker, yet only with regard to muscle or heart involvement [25].

## 2. Materials and Methods

A search through PubMed using as key words a variable combination of “myotonic dystrophy” with “CSF”, “CNS”, “blood-based biomarkers”, and “NfL” was used to select the research studies included in this mini review. Consensus statement articles were not included.

## 3. Results

### 3.1. CSF Biomarkers (Table 1)

Levels of total tau (T-tau), phosphorylated tau (P-tau), and the 42-amino-acid form of beta-amyloid (Ab42) in CSF were the first biomarkers studied in DM1 patients. They have been extensively studied in various neurodegenerative disorders: the highest increase of T-tau was found in Creutzfeldt–Jakob disease (CJD) and Alzheimer’s disease (AD), a mild-to-moderate increase in frontotemporal (FTD) and Lewy body dementia (LBD), while normal levels were found in Parkinson’s disease (PD) and progressive supranuclear palsy (PSP). Increased CSF levels of P-tau were found only in patients with AD, while a decrease in Ab42 was found in AD, amyotrophic lateral sclerosis (ALS), CJD, LBD, FTD, multiple system atrophy (MSA) and corticobasal syndrome (CBS) [26,27,28].

**Table 1 ijms-24-02204-t001:** Research studies concerning CSF biomarkers of neurodegeneration in DM1. Abbreviations: Ab40: 40-amino-acid form of beta-amyloid, Ab42: 42-amino-acid form of beta-amyloid, CSF: cerebrospinal fluid, DM1: myotonic dystrophy type 1, ELISA: enzyme-linked immunosorbent assay, MA: mean age, MDD: mean disease duration, P-tau: phosphorylated tau, T-tau: total tau, y: years.

Paper, Year	DM1 Patients	Controls	CSF Biomarkers Studied	Used Method
Winblad et al. [29], 2008	32, noncongenital (MA 41.8 y, MDD 16.8 y)	32 healthy controls (42.2 y)	Ab42 ↓ T-tau ↑ P-tau -	ELISA
Peric et al. [30], 2014	27 childhood/juvenile onset (MA 35 y, MDD 21.2 y) 47 adult onset (MA 46.7 y, MDD 17 y)	27 healthy controls (50.9 y)	Ab42 ↓ in juvenile-onset DM1 T-tau - P-tau - Ab42/P-tau ↓ in adult-onset DM1	ELISA
Laforce et al. [31], 2022	6 noncognitively impaired and 3 cognitively impaired DM1 patients	2 AD pts (67 y), no healthy controls	T-tau ↑ in 3 cognitively impaired DM1 P-tau ↑ in 3 cognitively impaired DM1 Ab42/Ab40 normal	automated chemiluminescent enzyme-immunoassay

The first study on CSF biomarkers in DM1 was conducted by Winblad et al., who found significant decrease of Ab42 and significant increase of T-tau by enzyme-linked immunosorbent assay (ELISA) in CSF of 32 noncongenital DM1 patients compared to controls [29]. No difference was found in P-tau levels, and no correlation between levels of CSF biomarkers and the number of CTG repeats was found. Unexpectedly, and differently from AD, they found a positive correlation between CSF T-tau and Ab42, with DM1 patients having either low Ab42 and normal T-tau or high T-tau and normal Ab42. Based on these results, the authors postulated the existence of two separate subgroups of DM1 patients characterized by two different CNS pathological processes. However, no differences were found between these two subgroups regarding any of the demographic or clinical features, possibly due to the limited sample size recruited [29].

Later on, Peric et al. studied the same CSF biomarkers by ELISA in a larger DM1 cohort, including 27 childhood/juvenile-onset DM1 patients, 47 adult-onset DM patients, and 27 controls, finding the lowest CSF Ab42 levels in the juvenile-onset DM1 group compared to the highest levels in controls [30]. Moreover, even if they did not find any significant differences in CSF levels of T-tau and P-tau among the three groups, the Ab42/P-tau ratio was decreased in adult-onset DM1 patients compared to controls. Based on these results, the authors suggested that reduced Ab42 levels could reflect the occurrence of a neurodevelopmental defect in juvenile-onset DM1 patients, while elevated tau protein levels, yet not reaching statistically significant results, may underlie a neurodegenerative process in adult-onset DM1 brains [30].

In a recent pilot study conducted on 12 DM1 and 2 AD patients, along with tau-PET imaging, Laforce et al. studied (i) CSF concentrations of Ab42, Ab40, T-tau, and P-tau by an automated chemiluminescent enzyme-immunoassay, (ii) plasma concentrations of Ab42 and Ab40 by a multiplex xMAP technique with a LABScan-200 system, and (iii) plasma concentrations of T-tau, Glial Fibrillary Acidic Protein (GFAP), and Neurofilament Light Chain (NfL) by single molecule array (SiMoA) [31]. No control cohort was included, only three out of 12 DM1 patients were cognitively impaired, and CSF studies were completed only in nine patients. Levels of CSF T-tau and P-tau were found to be higher in the three cognitively impaired DM1 patients compared to noncognitively impaired DM1 patients, but with a mean value lower than that typically observed in AD patients and more similar to other tauopathies such as FTD. As reported in the two previously cited studies [29,30], almost all DM1 participants had normal CSF Ab42/Ab40 ratio, suggesting the absence of AD-type central amyloidopathy [31]. This issue might be supported by the results of neuropathological studies on DM1 brains, showing the presence of neurofibrillary tangles but without the presence of amyloid plaques [13]. However, due to the very limited sample size, and since one of the DM1 patients enrolled may have had concomitant AD, drawing definite conclusions on these results may be misleading [31].

Overall, the results of these studies [29,30,31] indicate that CSF biomarkers are undoubtedly useful in helping to address the pathogenesis of CNS involvement in DM1. However, some major issues arise when proposing them as potential diagnostic and/or monitoring biomarkers to be used for clinical research purposes, especially in DM1. First, lumbar puncture (LP) is an invasive procedure, which may have some risks, especially for some categories of patients (i.e., DM1 patients on anticoagulant drugs for cardiac arrhythmias). Second, LP is painful; therefore, it should be reserved only for compelling diagnostic or therapeutic situations. Third, as cognitive and behavioral symptoms may significantly affect DM1 patients’ compliance to study protocols requiring to perform serial LPs, this might lead to bias selection in the studied population. For all these reasons, performing and repeating an LP over time is not a simple option to take into account, even in the setting of randomized controlled trials (RCTs).

Therefore, the search and validation of minimally invasive biomarkers is mandatory.

### 3.2. Blood-Based Biomarkers (Table 2)

The cognitive and behavioral disturbances that deeply affect DM1 patients’ compliance with medical prescriptions have pushed researchers to investigate the role of minimally invasive biomarkers that may be assessed over time with simple procedures, such as blood samples to obtain plasma or serum. Indeed, it has been shown in other neurodegenerative diseases that blood and CSF levels of some biomarkers are strongly related, while contrasting results have been obtained for others [32].

**Table 2 ijms-24-02204-t002:** Research studies concerning blood-based biomarkers of neurodegeneration in DM1. Abbreviations: Ab40: 40-amino-acid form of beta-amyloid, Ab42: 42-amino-acid form of beta-amyloid, AD: Alzheimer’s disease, CNS: central nervous system, CSF: cerebrospinal fluid, DM1: myotonic dystrophy type 1, ELISA: enzyme-linked immunosorbent assay, GFAP: Glial Fibrillary Acidic Protein, MA: mean age, MDD: mean disease duration, NfL: Neurofilament Light Chain, P-tau: phosphorylated tau, pts: patients, SiMoA: single molecule array, T-tau: total tau, y: years.

Paper, Year	DM1 Patients	Controls	Biomarkers Studied	Used Method
Saak et al. [33], 2021	9 DM1 pts (MA 44.3 y, MDD 16.8 y) out of 62 pts with myopathies with CNS involvement	485 healthy controls (MA 44.3 y), 13 pts with myopathies without CNS involvement (MA 41 y)	NfL ↑	SiMoA
van der Plas et al. [34], 2022	13 pre-manifest DM1 (MA 47.4 y) 40 DM1 pts (MA 46 y, MDD 12.9 y)	70 healthy controls (43.6 y)	NfL ↑ T-tau ↓ GFAP ↑ (only in pre-manifest DM1) UCH-L1 -	SiMoA
Nicoletti et al. [35], 2022	40 DM1 pts (MA 47.7 y, MDD 27.7 y)	22 healthy controls (MA 45.6 y)	NfL ↑	SiMoA
Laforce et al. [31], 2022	9 noncognitively impaired and 3 cognitively impaired DM1 patients (MA 47 y)	2 AD pts (MA 67 y), no healthy controls	NfL ↑ T-tau ↑ GFAP ↑ Ab42 ↑ Ab40 Ab42/Ab40 ratio ↑ (data regarding the 3 cognitively impaired DM1 pts vs. noncognitively impaired DM1 pts)	multiplex xMAP (Ab42, Ab40) SiMoA (NfL, T-tau, GFAP)

The first study that investigated blood-based biomarkers of CNS impairment in DM1 was performed by Saak et al., who assessed serum NfL by SiMoA in 62 patients with primary myopathies with known CNS involvement, 13 patients affected by myopathies without CNS impairment, and 8 patients with facioscapulohumeral dystrophy (FSHD) [33]. In the cohort of primary myopathies, DM1 patients only numbered nine, and their serum NfL levels were about two times higher compared to a previously gathered cohort of healthy controls, with a significant correlation between NfL levels and both age and n(CTG). Of note, in both DM1 and mitochondrial myopathy patients (*n* = 23), where more severe CNS damage was expected, on average, median NfL values were higher compared to the eight FSHD patients and 22 myotonic dystrophy type 2 patients (DM2). Interestingly, serum NfL significantly correlated with age only in patients affected by DM1 and DM2, supporting the idea that both forms of DM are progeroid disorders with premature aging [33].

A second study by van der Plas et al. [34] included 70 controls, 13 individuals with pre-manifest DM1 (PreDM1), and 40 patients with DM1, with the assessment of plasma levels of NfL, T-tau, GFAP, and ubiquitin C-terminal hydrolase-L1 (UCH-L1) by SiMoA. Regarding NfL, they found that PreDM1 and manifest DM1 had significantly higher NfL levels relative to controls. Moreover, they evaluated associations between NfL levels and both estimate progenitor allele length (ePAL) by small pool-PCR and cerebral white matter fractional anisotropy (WM FA) by 3T brain MRI. Interestingly, they found that patients with higher ePAL tended to have elevated NfL at a younger age relative to patients with lower ePAL [34]. Moreover, decreased FA was associated with increased NfL, making cerebral WM FA a significant predictor of NfL levels. Globally, the results of this study implied that plasma NfL is a sensitive marker of brain pathology in DM1 and could be useful in predicting disease onset. Nevertheless, the authors underlined that plasma NfL levels may not be suitable in tracking disease progression, as they and the other biomarkers studied did not change substantially over the period of the study. However, this statement should be cautiously taken, as manifest DM1 patients from this paper were mostly mildly affected with main Muscular Impairment Rating Scale (MIRS) score of two, not the whole DM1 disease-spectrum was then included. Most importantly, only 34 controls, 6 preDM1, and 27 DM1 patients completed the three scheduled visits over a period of 2 years.

With regard to the other biomarkers studied by van der Plas and colleagues [34], no differences among groups were found regarding UCH-L1, which is an enzyme of the ubiquitin–proteasome system, reported to be elevated in CNS trauma and some neurodegenerative diseases such as PD and AD. Regarding tau protein, they found significantly lower T-tau in DM1 patients compared to controls, and this is in contrast to data reported by the two CSF studies by Winblad et al. and Peric et al. [29,30]. The authors reported that this may be due to the lower amount of T-tau in blood compared to the CSF, or because SiMoA would be unable to detect the fetal tau isoform, which is mostly hyperphosphorylated and overexpressed in DM1 patients [34]. Lastly, and unexpectedly, GFAP, which is an intermediate filament protein expressed in astrocytes, was higher in the PreDM1 group compared to the other two, probably because there may be an increased reactivity of astrocytes in the first phase of the neurodegenerative process [34].

In our study [35], serum NfL levels were assessed by SiMoA in a cohort of 40 consecutive, adult DM1 patients vs. 22 healthy controls, finding that NfL levels were fourfold higher in DM1 patients. More than 90% patients (37/40) had NfL levels above the normal range, and, particularly, two out of three patients with the highest serum NfL values were congenital/infantile-onset forms (mean age 44.3 years), whereas two out of three patients with normal serum NfL showed normal cognitive performances (mean age 51.25 years). This supports the idea that NfL levels might actually reflect the global severity of cognitive impairment in DM1. Moreover, NfL levels significantly correlated with Fazekas score, which is a semiquantitative index of WM damage, and this reflects the results of van der Plas et al., who found a significant correlation between WM FA values and plasma NfL levels, therefore suggesting that that either structural or functional WM alterations might be the main basis of cognitive and behavioral alterations in DM1 patients [34,35].

Lastly, in the previously cited study by Laforce et al. [31], plasma biomarkers of neurodegeneration (NfL, Tau, GFAP, Ab42, and Ab42/Ab40 ratio) were all higher in the three cognitively impaired DM1 patients compared to the remaining noncognitively impaired DM1 patients (*n* = 9), while Ab40 plasma concentrations were similar between these two groups. Correlations between CSF and plasma biomarkers remained unclear (no *p*-values were reported), remarking a single significant direct correlation of plasma NfL levels with CSF P-tau concentration [31].

Globally, the results of the studies [31,33,34,35] on blood-based analytes in DM1 are quite consistent regarding the value of serum or plasma NfL as diagnostic biomarkers of CNS involvement. All the studies found elevated blood NfL when compared to controls, and the fact that blood NfL levels correlate with those in CSF should encourage further research in this direction. As a matter of fact, obtaining blood samples is definitely more simple, cost-effective, and safe than obtaining CSF. The use of blood samples should be, indeed, preferred also because it can be easily repeated over time, with better compliance from DM1 patients.

Regarding the other cited biomarkers, less consistent results have been obtained regarding tau protein detection in blood. In healthy brains, tau protein provides microtubule stability, facilitating intracellular trafficking, while in tauopathies, the normal function of tau is disrupted, ultimately leading to the development of NFTs [27]. DM1 is considered a tauopathy, as NFTs have been found in amygdala, hippocampus, entorhinal cortex, and temporal cortex of DM1 brains, with a topographic distribution similar to that reported for moderate AD, yet to a lesser extent, and this may be the first reason why data regarding tau in blood from DM1 patients are less robust [13]. Indeed, the concentration of tau in plasma is approximately 100-fold lower than that in CSF, and, given the limited amount of tau aggregates in brains from DM1 patients, it is possible that even tools such as SiMoA might not be sensitive enough to detect extremely small amounts of this analyte. A similar speculation may be made for Ab40/Ab42, with the further challenge related to the fact that both plasma and serum have higher total protein concentration with a more complex protein matrix than CSF, that additionally reduces the concentration of blood Ab proteins available for measurement [36].

Moreover, the tau protein has six tau isoforms, which differ in the inclusion or exclusion of two amino-terminal exons 2 and 3 (0N, 1N, or 2N) and contain three to four amino acid repeat sequences in the microtubule-binding domain encoded by exon 10 of *MAPT* (3R and 4R, respectively) [37]. In the brain, alterations in the expression of different tau isoforms and their aggregation have been linked to the most common tauopathies, with PSP and CBS showing 4R tau-containing inclusions, Pick’s disease expressing filamentous 3R tau, and AD NFTs containing both 3R and 4R forms [38]. Compared to AD brains, DM1 is characterized by the prevalent aggregation of the 0N3R isoform, which lacks the exons 2 and 3. In addition to exons 2 and 3, mis-splicing of tau has also been reported for exon 10, although to a lesser extent, and even less so for exon 6 [39]. In diverse tauopathies, different tau isoforms can be found in the CSF, while in blood the dominant isoform is the full-length one, probably reflecting the fact that T-tau found in blood mainly derives from peripheral sources [38]. Therefore, the commercially available SiMoA T-tau assay, which measures mid regions of all tau protein isoforms, in blood may perform worse than in the CSF.

To complicate the matter, the tau protein undergoes various post-translational modifications, such as hyperphosphorylation, acetylation, and N-glycosylation, and, in particular, P-tau has been extensively studied in AD [37]. The phosphorylation of tau is a process regulated during brain development, with fetal tau more phosphorylated than the normal adult one, and further increase in tau phosphorylation in AD has been described. There are 85 potential phosphorylation sites in the full-length tau, and a specific SiMoA assay should be designed to recognize each of them [40]. As the degree of phosphorylation of tau in DM1 is still unknown, it is possible that it has a different pattern from AD, requiring further research before using it as a biomarker of CNS degeneration in DM1.

## 4. Concluding Remarks

Data in the current literature support, in particular, NfL as a potential diagnostic biomarker of CNS involvement in DM1 as well.

Further cross-sectional data from larger DM1 cohorts stratified according to the severity of cognitive/behavioral involvement are needed to assess its reliability as a predictive and monitoring biomarker of CNS involvement. In this regard, NfL is already recognized as a sensitive diagnostic, monitoring, prognostic, and predictive biomarker of neuroaxonal damage in many other neurological conditions (i.e., multiple sclerosis), a circumstance that should ensure to shortly fill the gap regarding the need for standardization of blood sample collection and NfL measurement to be applied in different laboratories. This issue appears particularly relevant in the case of rare diseases such as DM1, because it could facilitate enrollment of large study cohorts through a collaborative research effort among reference tertiary centers.

Finally, the availability of standardized blood-based biomarkers of CNS disease severity will also simplify designing of future DM1 clinical trials aiming to evaluate the effects of disease-modifying treatments also tailored for CNS involvement.

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
