# Peer review of "Fluid Biomarkers of Central Nervous System (CNS) Involvement in Myotonic Dystrophy Type 1 (DM1)"

_ijms, 2023, doi:10.3390/ijms24032204_

Round 1

Reviewer 1 Report

The authors discuss biomarkers of central nervous system involvement in Myotonic Dystrophy type 1 in a mini-review. The need for CNS biomarkers in MD1 has been voiced by the DM-CNS group in 2015 (Bosco et al. 2015, not cited by the authors).

They limit their discussion to biomarkers from body fluids, although this is not obvious from the title of the MS.

Their presentation of pathophysiology of MD1 is adequate.

Unfortunately, the FDA typology of biomarkers (susceptibility/risk, diagnostic, monitoring, prognostic, predictive, pharmacodynamic/response, and safety) is not used. Therefore, it remains unclear what clinical functions CNS biomarkers from body fluids are meant to fulfil according to the authors.

Numerous imaging studies aimed at biomarker discovery are mentioned in only a perfunctory manner.

There are several studies on miRNA as biomarkers in DM1, which were omitted by the authors. Besides, DAPK1 has been proposed recently as a possible „CNS biomarker“ in DM1.

Some of the cited studies suggest an increase in T-tau in CSF or a decrease in blood. The role of amyloid-β as a biomarker remains dubious.

According to the authors, current studies consistently show that NFL represents a potential biomarker of CNS pathology in DM1. We do not concur with this view, as the largest study by Saak et al. showed rather similar median NFL levels in serum in DM1, DM2 and FSHD; interestingly, FSHD represents a disease that is not characterized by a prominent CNS pathology. The overlap with NFL levels of controls was considerable in this study. In our view, it might thus become possible to detect an NFL increase on a group level, but will probably prove difficult on an individual basis. Furthermore, an age-dependency of NFL levels should not be disregarded. Longitudinal studies on fluid biomarkers in DM1 are missing.

Please check language and spelling (e.g., line 22 with; line 72 manifests; line 75 eliminate „main“; line 76 neuropathological; line 129 through; no publication year ref. 3).

Author Response

We thank the reviewer for his/her helpful comments. Please find below a point-to-point answer to his/her concerns.

The authors discuss biomarkers of central nervous system involvement in Myotonic Dystrophy type 1 in a mini-review. The need for CNS biomarkers in MD1 has been voiced by the DM-CNS group in 2015 (Bosco et al. 2015, not cited by the authors).

Answer: Actually, the reference you are referring to was present in the first submission with the number 36, but we forgot to add the corresponding voice in the main text. We corrected it (current ref. n°24).

They limit their discussion to biomarkers from body fluids, although this is not obvious from the title of the MS.

A: We changed the title accordingly.

Their presentation of pathophysiology of MD1 is adequate.

Unfortunately, the FDA typology of biomarkers (susceptibility/risk, diagnostic, monitoring, prognostic, predictive, pharmacodynamic/response, and safety) is not used. Therefore, it remains unclear what clinical functions CNS biomarkers from body fluids are meant to fulfil according to the authors.

A: We changed this, using the FDA nomenclature throughout the text.

Numerous imaging studies aimed at biomarker discovery are mentioned in only a perfunctory manner.

A: In the introduction we specified that we only mentioned few longitudinal imaging studies as there was already an extensive, recent and well structured review (Minnerop M et, Current Progress in CNS Imaging of Myotonic Dystrophy. Front Neurol. 2018) on neuroimaging biomarkers in DM1, that we cited for reference about this topic.

There are several studies on miRNA as biomarkers in DM1, which were omitted by the authors. Besides, DAPK1 has been proposed recently as a possible „CNS biomarker“ in DM1.

A: Although miRNA have been considered as potential biomarkers, no study on miRNAs in DM1 was focused on CNS involvement. We added a sentence and a reference to explain this. Also the study on DAPK1 that you mentioned proposed it as a global disease biomarkers, not specific for CNS.

Some of the cited studies suggest an increase in T-tau in CSF or a decrease in blood. The role of amyloid-β as a biomarker remains dubious.

According to the authors, current studies consistently show that NFL represents a potential biomarker of CNS pathology in DM1. We do not concur with this view, as the largest study by Saak et al. showed rather similar median NFL levels in serum in DM1, DM2 and FSHD; interestingly, FSHD represents a disease that is not characterized by a prominent CNS pathology. The overlap with NFL levels of controls was considerable in this study. In our view, it might thus become possible to detect an NFL increase on a group level, but will probably prove difficult on an individual basis. Furthermore, an age-dependency of NFL levels should not be disregarded. Longitudinal studies on fluid biomarkers in DM1 are missing.

A: We added the age at the time of the study for both patients and controls. Indeed, the study by Saak et al. included only 9 patients with DM1 and only 8 patients with FSHD. Considering the small sample size and correcting by the age, the authors themselves verified that compared to the controls, serum NfL levels were above the 90th percentile in 6/9 (66%) patients with DM I and in only 3/8 (38%) FSHD patients. Furthermore, FSHD patients had mean age of 49.4 years compared to 44.3 years of DM1 patients.

In order to give to the readers information about age range of patients and controls, we included it in the tables summarizing the studies.

Please check language and spelling (e.g., line 22 with; line 72 manifests; line 75 eliminate „main“; line 76 neuropathological; line 129 through; no publication year ref. 3).

A: We changed them.

Reviewer 2 Report

The manuscript is a comprehensive review of the usefulness of certain biomerkers in Myotonic Dystrophy Tye I. cases.  The suggested Neurofilament Light Chain as biomarker semms to be relevant both in CSF and blood samples, with superior fetures over tau nd beta amyloid porteins. The standardization and interpretation requirements should be fullfilled before the test can be recommened as a biomarker in routine diagnostic and prognostic protocols. The manuscript is clear and straightforward,  with moderate conclusions as an advantage in this sense. It can be recommended for publication in the present form.

Author Response

We really thank you for your comment.

Reviewer 3 Report

In this manuscript by Rossi and Silvestri, the authors present a systemic review to establish CSF and blood-based biomarkers to measure CNS involvement in myotonic dystrophy type 1 (DM1). Such biomarkers are highly needed to allow future clinical trials to evaluate the effects of treatments on the CNS in DM1. Therefore, the paper is relevant and potentially useful for clinical research. The manuscript is also well written and structured. However, I have some comments to be considered.

The age is an important factor potentially affecting Tau and NfL in the CSF and the blood. More details about the age of the patients and the controls should be added, especially in the section 3.1. Given that the symptoms in adult DM1 are proposed to be from a neurodegenerative cause, the duration of the symptoms of adult DM1 patients when they are sampled should be discussed, when possible. For instance, a patient with adult-onset DM1 who manifested the symptoms at 20 years old is expected to have different biomarkers levels than a patient who developed the symptoms after 35 years old.

The authors state that that NfL reflect the global severity of cognitive impairment in DM1 because of higher NfL level in the blood of early-onset DM1 patients when compared to patient with adult-onset DM1 (lines 248 to 242). What was the age of the DM1 in the cited study? Cognitive impairments are often present from childhood in patients with early-onset DM1, but I doubt that NfL or Tau levels are significatively affected at such a young age when the signs of neurodegeneration are absent. 

In the lines 252-256, the correlation between the NfL levels and white matter damages do not directly support that white matter damages are the main basis of cognitive and behavioral alterations, as stated by the author. This kind of conclusion would be better supported if comparing with data from neuropsychological tests and cognitive evaluation. 

I also have some minor corrections:

-Abstract, line 22 :  (…) of CNS involvement, whith less robust results regarding (…).

-Introduction, line 49 : (…) from the mutant DMPK allele, (…).

-Introduction, ligne 118 : Define WM and GM.

-Introduction, lines 123-127 : Revise punctuation .

-Materials and methods, line 129 : A search throughh Pubmed (…)

-Materials and methods, line 131 : unnecessary space in (…) included in this mini review .

-Materials and methods, lines 131-132 : Consensus statement articles which were not included.

-Results, line 156, ELISA could be defined when used for the first time in line 145.

R-esults, line 166 : (…) Laforce et al. studied (…).

-Results, line 168 : Both the abbreviation Aβ1-42 and Ab42 are used. For consistency, only one form should be used.

-The role of both tau isoforms in neurodegenerative diseases could be briefly described for readers unfamiliar with the subject.

-Results, line 204 : (…) by Saak et al., who assessed (…).

-Results, line 204 : Both the abbreviations SiMoA and Simoa are used. For consistency, only one form should be used.

-Results, line 207 : (…) [28]. In the cohort (…).

-Results, line 228 : Is the comma necessary in the following sentence? (…) studied, did not change (…).

-Results, line 261 : (…) DM1 patients (n=9), while (…)

-Results, line 296 : (…) than in the CSF (…).

-Table 1 and 2 : Maybe the abbreviation pts could be define somewhere

T-able 1 and 2  : Biomarkers studied

-References, line 418 : The reference 35 seems to not be formatted the same way as the others.

Author Response

We thank the reviewer for his/her helpful comments. Please find below a point-to-point answer to his/her concerns.

In this manuscript by Rossi and Silvestri, the authors present a systemic review to establish CSF and blood-based biomarkers to measure CNS involvement in myotonic dystrophy type 1 (DM1). Such biomarkers are highly needed to allow future clinical trials to evaluate the effects of treatments on the CNS in DM1. Therefore, the paper is relevant and potentially useful for clinical research. The manuscript is also well written and structured. However, I have some comments to be considered.

The age is an important factor potentially affecting Tau and NfL in the CSF and the blood. More details about the age of the patients and the controls should be added, especially in the section 3.1. Given that the symptoms in adult DM1 are proposed to be from a neurodegenerative cause, the duration of the symptoms of adult DM1 patients when they are sampled should be discussed, when possible. For instance, a patient with adult-onset DM1 who manifested the symptoms at 20 years old is expected to have different biomarkers levels than a patient who developed the symptoms after 35 years old.

Answer: We added the age of patients and controls in the tables, and we also added the disease duration, when available. This is for sure a good point, but we should also consider that: 1) disease duration in all papers was evaluated taking into account the first muscular symptom and not cognitive/behavioural features, especially for adult-onset forms; 2) pathogenesis of cognitive abnormalities in DM1 is a matter of debate, as neurodevelopmental, neurofunctional, and neurodegenerative hypotheses have been variously documented by both neuroimaging and neuropathological studies in different disease forms (e.g., congenital vs adult-onset vs late-onset forms). Further longitudinal studies are definitely needed, with a better stratification of patients.

The authors state that that NfL reflect the global severity of cognitive impairment in DM1 because of higher NfL level in the blood of early-onset DM1 patients when compared to patient with adult-onset DM1 (lines 248 to 242). What was the age of the DM1 in the cited study? Cognitive impairments are often present from childhood in patients with early-onset DM1, but I doubt that NfL or Tau levels are significatively affected at such a young age when the signs of neurodegeneration are absent. 

A: We added also the mean age at the time of the study of the 3 congenital/infantile-onset forms (44,3 years) and the 3 patients with normal serum NfL (51,25 years). As previously stated, a neurodegenerative process superimposed to the neurodevelopmental defect may occur in patients with congenital forms.       

In the lines 252-256, the correlation between the NfL levels and white matter damages do not directly support that white matter damages are the main basis of cognitive and behavioral alterations, as stated by the author. This kind of conclusion would be better supported if comparing with data from neuropsychological tests and cognitive evaluation. 

A: Van der Plaas et al. studied white matter fractional anisotropy in DM1, which is known to be globally impaired rather than tract-specific in these patients (Koscik TR, van der Plas E, Gutmann L, Cumming SA, Monckton DG, Magnotta V, et al. White matter microstructure relates to motor outcomes in myotonic dystrophy type 1 independently of disease duration and genetic burden. Sci Rep. 2021). They found that Cerebral WM FA was a significant predictor of NfL, where decreased FA was associated with increased NfL. In our paper (Nicoletti et al), we only used a less specific marker of white matter dysfunction, namely the Fazekas score assessed on standard brain MRI, which positively correlated with blood NfL levels. Studies published by Serra et al. and Minnerop et. al documented that white matter involvement might be at the basis of CNS manifestations in DM1 (i.e., Minnerop M, Weber B, Schoene-Bake JC, Roeske S, Mirbach S, Anspach C, Schneider-Gold C, Betz RC, Helmstaedter C, Tittgemeyer M, Klockgether T, Kornblum C. The brain in myotonic dystrophy 1 and 2: evidence for a predominant white matter disease. Brain. 2011 Dec;134(Pt 12):3530-46. doi: 10.1093/brain/awr299. Epub 2011 Nov 29. PMID: 22131273; PMCID: PMC3235566). However, we changed the sentence, to make less strong conclusions.

I also have some minor corrections:

-Abstract, line 22 :  (…) of CNS involvement, whith less robust results regarding (…).

A: We changed it

-Introduction, line 49 : (…) from the mutant DMPK allele, (…).

A: We changed it

-Introduction, ligne 118 : Define WM and GM.

A: We defined them.

-Introduction, lines 123-127 : Revise punctuation .

We revised it

-Materials and methods, line 129 : A search throughh Pubmed (…)

A: We changed it

-Materials and methods, line 131 : unnecessary space in (…) included in this mini review .

A: We removed it.

-Materials and methods, lines 131-132 : Consensus statement articles which were not included.

A: We removed it.

-Results, line 156, ELISA could be defined when used for the first time in line 145.

A: We defined it the first time it appeared.

R-esults, line 166 : (…) Laforce et al. studied (…).

A: We added the period.

-Results, line 168 : Both the abbreviation Aβ1-42 and Ab42 are used. For consistency, only one form should be used.

A: We modified the text using always the same abbreviation.

-The role of both tau isoforms in neurodegenerative diseases could be briefly described for readers unfamiliar with the subject.

A: We added it.

-Results, line 204 : (…) by Saak et al., who assessed (…).

A: We added it.

-Results, line 204 : Both the abbreviations SiMoA and Simoa are used. For consistency, only one form should be used.

A: We changed it.

-Results, line 207 : (…) [28]. In the cohort (…).

A: We changed it.

-Results, line 228 : Is the comma necessary in the following sentence? (…) studied, did not change (…).

A: We removed it.

-Results, line 261 : (…) DM1 patients (n=9), while (…)

A: We modified it.

-Results, line 296 : (…) than in the CSF (…).

A: We modified it.

-Table 1 and 2 : Maybe the abbreviation pts could be define somewhere

A: We specified it.

T-able 1 and 2  : Biomarkers studied

A: We changed it.

-References, line 418 : The reference 35 seems to not be formatted the same way as the others.

A: We changed it.